# An sEMG-Controlled Forearm Bracelet for Assessing and Training Manual Dexterity in Rehabilitation: A Systematic Review

**DOI:** 10.3390/jcm11113119

**Published:** 2022-05-31

**Authors:** Selena Marcos-Antón, María Dolores Gor-García-Fogeda, Roberto Cano-de-la-Cuerda

**Affiliations:** 1International Doctorate School, Rey Juan Carlos University, 28008 Madrid, Spain; s.marcosa@alumnos.urjc.es; 2Department of Physical Therapy, Occupational Therapy, Rehabilitation and Physical Medicine, Rey Juan Carlos University, 28922 Alcorcon, Spain; mariadolores.gor@urjc.es

**Keywords:** activities of daily living, dexterity, functional independence, MYO armband, rehabilitation, semi-immersive virtual reality, technologies, upper limb impairment, virtual reality

## Abstract

Background: The ability to perform activities of daily living (ADL) is essential to preserving functional independence and quality of life. In recent years, rehabilitation strategies based on new technologies, such as MYO Armband^®^, have been implemented to improve dexterity in people with upper limb impairment. Over the last few years, many studies have been published focusing on the accuracy of the MYO Armband^®^ to capture electromyographic and inertial data, as well as the clinical effects of using it as a rehabilitation tool in people with loss of upper limb function. Nevertheless, to our knowledge, there has been no systematic review of this subject. Methods: A systematically comprehensive literature search was conducted in order to identify original studies that answered the PICO question (patient/population, intervention, comparison, and outcome): What is the accuracy level and the clinical effects of the MYO Armband^®^ in people with motor impairment of the upper limb compared with other assessment techniques or interventions or no intervention whatsoever? The following data sources were used: Pubmed, Scopus, Web of Science, ScienceDirect, Physiotherapy Evidence Database, and the Cochrane Library. After identifying the eligible articles, a cross-search of their references was also completed for additional studies. The following data were extracted from the papers: study design, disease or condition, intervention, sample, dosage, outcome measures or data collection procedure and data analysis and results. The authors independently collected these data following the CONSORT 2010 statement when possible, and eventually reached a consensus on the extracted data, resolving disagreements through discussion. To assess the methodological quality of papers included, the tool for the critical appraisal of epidemiological cross-sectional studies was used, since only case series studies were identified after the search. Additionally, the articles were classified according to the levels of evidence and grades of recommendation for diagnosis studies established by the Oxford Center for Evidence-Based Medicine. Also, The Cochrane Handbook for Systematic Reviews of Interventions was used by two independent reviewers to assess risk of bias, assessing the six different domains. The Preferred Reporting Items for Systematic Reviews and Meta-Analyses (PRISMA) was followed to carry out this review. Results: 10 articles with a total 180 participants were included in the review. The characteristics of included studies, sample and intervention characteristics, outcome measures, the accuracy of the system and effects of the interventions and the assessment of methodological quality of the studies and risk of bias are shown. Conclusions: Therapy with the MYO Armband^®^ has shown clinical changes in range of motion, dexterity, performance, functionality and satisfaction. It has also proven to be an accurate system to capture signals from the forearm muscles in people with motor impairment of the upper limb. However, further research should be conducted using bigger samples, well-defined protocols, comparing with control groups or comparing with other assessment or therapeutic tools, since the studies published so far present a high risk of bias and low level of evidence and grade of recommendation.

## 1. Introduction

The ability to perform activities of daily living (ADL) is essential to preserve functional independence and quality of life [1]. This ADL performance can be severely restricted in people with neurological disorders such as children with congenital disorders or developmental disabilities and adults with acquired injuries or neurodegenerative diseases, due to upper limb motor impairment [1]. These functional disorders are often related to loss of dexterity, which is defined as “fine, voluntary movements used to manipulate small objects during a specific task” [2]. Also, dexterity is associated with two related concepts: manual dexterity (ability to handle objects with the hand) and fine motor dexterity (in-hand manipulations as separate skills from the gross motor grasp and release skills associated with manual dexterity) [2]. Hence, rehabilitation processes should enable patients to restore their functional capacity by training dexterity [3].

In recent years, rehabilitation strategies based on new technologies have been implemented to improve dexterity in people with upper limb impairment, enhancing patient comfort [3]. Virtual reality (VR) seems to have the potential to improve upper limb rehabilitation [4], since it creates virtual environments similar to the real world where the user can interact with appealing surroundings and perform significant, high repetition, high transfer capacity, and motivating tasks, maximizing neuroplasticity and motor learning thanks to the provided feedback [5,6]. Also, these motivating and appealing environments created by VR could help recover health conditions and community integration [7]. Furthermore, these devices achieve higher intensity at a sustainable cost [5,6,8].

The MYO Armband^®^ is a semi-immersive VR device that captures forearm movements [9]. It consists of an accelerometer, a gyroscope, a magnetometer (inertial measure unit (IMU)) and eight surface electromyography (sEMG) sensors [9,10]. The electromyographic signal is streamed wirelessly at a frequency of 200 Hz and the orientation and position data from the inertial sensor is transmitted at a frequency of 50 Hz. This system, which integrates motion tracking and electromyography with VR, provides quantitative data on muscle activity that can be used not only for objective assessment but also as a semi-immersive VR therapeutic tool [9,10].

Over the last few years, many studies have been published focusing on the accuracy of the MYO Armband^®^ to capture electromyographic and inertial data [4,11], as well as the clinical effects of using it as a rehabilitation tool in people with loss of upper limb function [8,9,10]. Nevertheless, to our knowledge, there has been no systematic review of this subject. Therefore, we conducted a systematic review with the aim of analyzing the accuracy and the clinical effects of using the MYO Armband^®^ in people with motor impairment of the upper extremity.

## 2. Materials and Methods

### 2.1. Design

The Preferred Reporting Items for Systematic Reviews and Meta-Analyses (PRISMA) [12] was used to carry out this systematic review, starting with a PICO (patient/population, intervention, comparison, and outcome) question: Which is the accuracy level and the clinical effects of the MYO Armband^®^ in people with motor impairment of the upper limb as compared with other assessment techniques or interventions or no intervention whatsoever?

### 2.2. Search Strategy

A systematically comprehensive literature search was conducted from June to November 2021 in order to identify original studies that answered the PICO question, using the following data sources: Pubmed, Scopus, Web Of Science (WOS), ScienceDirect, Physiotherapy Evidence Database (PEDro) and the Cochrane Library. After identifying the eligible articles, a cross-search of their references was also completed for additional studies.

The combinations of keywords were: *“MYO Armband” AND (rehabilitation OR “manual dexterity” OR “upper limb” OR disability)*. The detailed search strategy for each database is shown in Table 1.

Two authors independently searched and screened titles and abstracts to identify studies meeting inclusion criteria. Duplicates were removed and disagreements regarding the selection of studies were resolved by a third author.

### 2.3. Study Selection

Studies published in Spanish and English between January 2016 and November 2021 were considered for inclusion in this review, regardless of their methodological design.

The exclusion criteria were: no access to full-text, poster communications, congress or symposium reports, and technical analysis studies with no clinical application or perspective.

### 2.4. Participants

This review considered studies that included subjects with motor impairment of the upper limb. Studies that included healthy subjects as the control group (CG) and studies that analyzed the accuracy of the device comparing healthy subjects with affected subjects were also taken into consideration.

### 2.5. Interventions

For the clinical trials, the intervention group had to follow a rehabilitation program using the MYO Armband^®^ either isolated or combined with other therapeutic strategies, in any dosage and provided in any setting (inpatient, outpatient, or domicile).

For the case studies, they had to analyze the accuracy of the MYO Armband^®^ or assess and/or carry out an intervention using sEMG in people with motor impairment of the upper limb.

### 2.6. Outcome Measures

Studies that analyzed parameters related to the accuracy to capture signals and the functioning of the system were included. Additionally, studies that analyzed outcomes that measure mobility, dexterity, and upper limb function, as well as outcomes related to these parameters were also included.

### 2.7. Data Extraction and Analysis

The following data were extracted from the papers: study design, disease or condition, intervention, sample, dosage, outcome measures or data collection procedure, and data analysis and results. The authors independently collected these data following the CONSORT 2010 statement [13] when possible, and eventually reached a consensus on the extracted data, resolving disagreements through discussion.

### 2.8. Assessment of Methodological Quality of the Studies and Risk of Bias

To assess methodological quality, we used the tool for the critical appraisal of epidemiological cross-sectional studies adapted by Ciaponni [14] from the work of Berra et al. [15], since only case series studies were identified after the search. This tool contains 31 items divided into 6 dimensions: (a) research question or aim of the research, (b) participants, (c) comparability between groups, (d) definition and measurement of the primary variables, (e) statistical analysis and confusion, (f) results, (g) conclusions, external validity and applicability of the results, (h) conflict of interests and (i) follow-up. Dimensions “b” to “e” assess internal validity and items 25 and 26 assess external validity. Methodological quality is considered high when most of the dimensions are rated as “good” or “very good”; medium when most of the dimensions are rated as “good” or “fair” or internal validity is rated as “medium”; and low when most of the dimensions are rated as “fair” or “bad” or internal validity is rated as “low”.

Additionally, the articles were classified according to the levels of evidence and grades of recommendation for diagnosis studies established by the Oxford Center for Evidence-Based Medicine [16].

The Cochrane Handbook for Systematic Reviews of Interventions [17] was used by two independent reviewers to assess the risk of bias, assessing the six different domains:(a)Selection bias: relates to recruiting process and participant allocation. To analyze it, randomization and allocation concealing must be considered.(b)Performance bias: refers to systematic differences between groups in the care that is provided, or in exposure to factors other than the interventions of interest. To analyze it, blinding procedures must be examined.(c)Detection bias: refers to systematic differences between groups in how outcomes are determined and may occur during intervention and follow-up. Blinding of outcome assessors must be considered when analyzing it, since it may reduce the risk.(d)Attrition bias: systematic differences between groups in withdrawals from a study. It occurs when there are withdrawals that lead to incomplete outcome data or when withdrawals in both groups differ significantly.(e)Reporting bias: refers to systematic differences between reported and unreported findings. This can occur once the study is finished and it is due to the selective report of results, reporting only statistically significant data.(f)Other biases: occur when reviewers include methodological aspects that are not assessed in the domains described before. They relate mainly to certain trial designs, such as crossover trials.

Each study was assessed independently and was considered a “low risk of bias” when each domain was addressed properly. Otherwise, it was considered a “high risk of bias”. If a study did not provide enough information, it was considered “dubious”. Disagreement was resolved through discussion with a third reviewer.

## 3. Results

The literature search and the article selection process are detailed in Figure 1. The initial search yielded 108 articles. Once the duplicates were removed and eligibility criteria were applied, 10 articles with a total of 180 participants were included in the review [4,7,9,10,11,18,19,20,21,22]. The characteristics of included studies are shown in Table 2 and Table 3.

### 3.1. Sample Characteristics

One study (n = 19) included a sample of children with cerebral palsy (CP) with ages ranging from 8 to 18 years [22]. Three studies (n = 7) included patients with upper limb amputation: one of them did not specify the amputation level [4], another included transradial amputees [19] and the other transhumeral amputees [20]. Five studies (n = 31) examined patients with stroke [7,9,11,18,21]: three of them did not specify the characteristics of the impairment, one included people (n = 3) with mildly to severely impaired hand function [11], and the other included people (n = 4) with different levels of impairment [21]. Two studies examined people (n = 49) with multiple sclerosis (MS) [9,10] but did not specify the level of disability in the Expanded Disability Status Scale (EDSS) and one study included patients with traumatic brain injury (TBI) without specifying cause or severity [9]. Combined, these articles included 57 healthy subjects to compare their results with those of the patients who presented some condition or disease.

### 3.2. Intervention Characteristics

All studies used semi-immersive VR. Six of them [4,11,18,19,20,21] implemented protocols designed to analyze the accuracy of the sEMG system. One of these used a protocol based on a dance game and added the Kinect^®^ sensor to track position in amputees, with the aim of knowing the time taken for each hand gesture to be detected by the system as well as its total operating time [4]. Another compared the accuracy of the information captured by the MYO Armband^®^ with functional near-infrared spectroscopy (fNIRS) and evaluated the effectiveness of the combination of both systems to obtain information about motor intention in patients with transhumeral amputation, which may be useful to improve the control of upper limb prostheses [20]. The other four studies [11,18,19,21] used different upper limb exercise protocols with or without visual feedback that allowed to calculate the accuracy of the MYO Armaband^®^.

The rest of the articles included in this review [7,9,10,22] analyzed the clinical effects of video game-based therapy with the MYO Armband^®^. Three of them [7,9,10] combined the use of the MYO Armband^®^ with other devices such as the Kinect^®^ sensor or a foot pedal for gaming.

The studies were heterogeneous regarding dosage. Processes to obtain data in those studies analyzing the sensor’s accuracy differed significantly. In the studies evaluating the clinical effects of the MYO Armband^®^ combined with semi-immersive VR, the mean session duration was 45.66 ± 24.82 min (range 17–60 min) [7,9,10,22]; only one study specified the number of sessions (50 sessions) [10]; and the mean number of weeks was 7.33 ± 3.05 (range 4–10 weeks) [7,10,22].

### 3.3. Outcome Measures

The studies that examined the accuracy to capture sEMG signals used different classification and data processing algorithms, as well as the information provided by the sEMG in Hz. Melero et al. [4] evaluated the operating time, which is the time taken for each hand gesture to be detected by the system from the moment it appears on the screen. It can be broken down into detection time (the time it takes for each specific gesture to be recognized by the system) and reaction time (the time it takes the subject to perform a hand gesture from the moment it appears on the screen). In addition, three studies [11,19,21] analyzed the accuracy of the MYO Armband^®^ by using classification algorithms in people with stroke and in transradial amputees. They used a software to apply mathematical formulations to the data collected from the sEMG, calculating the percentages of the accuracy of the system. Finally, two studies [18,20] used the sEMG signal to know the characteristics of the forearm muscle contraction performed by the participants. The second [20] also compared this signal with the data collected from another motion capture device and analyzed the effectiveness of combining both devices as an assessment approach.

The studies were also heterogeneous regarding outcome measures. Those articles analyzing clinical effects used physical evaluation tools, as well as functional and cognitive. One of the articles [22] used the Assisting Hand Assessment (AHA) to assess spontaneous bimanual performance and the Box and Blocks Test (BBT) to assess unilateral hand dexterity. Two articles [7,10] used the Motor Assessment Scale (MAS), a scale to assess motor function in people with stroke. They also evaluated wrist extension, grip strength, and angular velocity [7,22]. Range of motion (ROM) was also assessed in most of the studies. In relation to functional and cognitive assessment, one study [22] used the Canadian Occupational Performance Measure (COPM) to assess self-perception of performance in everyday living and the Self-Reported Experiences of Activity Settings (SEAS) to assess participation experiences. Finally, another study [7] assessed the patient’s engagement during therapy with the Engagement Questionnaire (EQ).

### 3.4. Accuracy of the System and Effects of the Interventions

In regard to the accuracy of the system to assess motor control of the upper limb, we found values that varied between 78% and 99% in mildly to severely impaired subjects after stroke [11,21], and the accuracy for three gestures involved in ADL (rest, close, open, key pinch and precision pinch) was 94% [11]. On the other hand, no event was reported regarding calibration, donning, or executing tasks with the device [18]. In patients with amputation, the results showed that it takes less than 3 s on average for each gesture to be detected and an overall operating time below 4 s. Since the authors expected the operating time to be below 6 s, they concluded that the MYO Armband^®^ was suitable for accurately detecting gestures in people with amputations of the upper limb [4]. Additionally, the study that included transradial amputees [19] found a percentage of accuracy of 93.3% whereas the one that included transhumeral amputees [20] reported an accurfor of 94.6% and 74% for elbow and wrist movements respectively. This accuracy increased significantly by combining sEMG with fNIRS, which demonstrates the feasibility of a hybrid sEMG and fNIRS system to improve the control performances of multifunctional upper-limb prostheses.

Secondly, those studies that examined clinical effects showed improvements after intervention with the MYO Armband^®^. One study found improvements in functionality related to ROM [10], whereas another showed a moderate increase in grip strength and dexterity as well as higher scores in the COPM and SEAS [22]. Also, some studies reported that the participants showed high interest and engagement during the activities due to a good feeling of immersion in the game, as well as a feeling of enjoyment and motivation [7,8,10].

### 3.5. Assessment of Methodological Quality of the Studies and Risk of Bias

Table 4 shows the results obtained after analyzing the quality of the studies using the tool for the critical appraisal of epidemiological cross-sectional studies [14]. Internal validity was rated as low for 60% of the articles, medium in 30%, and high in 10% of them. External validity was poor for 100% of the studies. Overall methodological quality was rated as low for 60% of the articles, medium for 30%, and high for 10%.

The levels of evidence and grades of recommendation are detailed in Table 5. All articles were classified as level of evidence 4, with a grade of recommendation of C.

Figure 2 summarizes the results of the assessment of the risk of bias sorted by article.

## 4. Discussion

For years, information and communication technologies have been used as an assessment and therapeutic tool in the field of rehabilitation [23,24]. VR-based therapy has been increasingly implemented to complement conventional therapy in people with motor control disorders [25,26]. This is, to our knowledge, the first systematic review that summarizes the available evidence on the use of the MYO Armband^®^ as an assessment and rehabilitation tool in people with motor impairment of the upper limb. Our results suggest that the use of the MYO Armband^®^ as a therapeutic tool in people with motor impairment of the upper limb improves ROM, grip strength, dexterity, functionality, and ADL performance. They also show good satisfaction and feeling of immersion reported by users. Furthermore, the MYO Armband^®^ could be considered an assessment tool suitable to detect small changes during the rehabilitation progress, since the accuracy of the system proved to be high.

Semi-immersive VR combined with the MYO Armaband^®^ provide opportunities for motor and cognitive tasks recreating real-life scenarios and simulations of activities by capturing human motion [27]. Also, semi-immersive VR has shown to have fewer side effects, such as “cybersickness”, compared to immersive VR [27,28]. For these reasons, together with the wide availability of these systems and their sustainable cost, semi-immersive VR is recommended to complement conventional therapy in the rehabilitation of upper limb impairment [27].

In the articles included in this review, the MYO Armband^®^ was combined with different devices. On one hand, some studies used these combinations in order to obtain more information about the orientation, position, and movement of the upper limb. For instance, Esfalahni et al. [7,9,10] combined the MYO Armband^®^ with the Kinect^®^ sensor to increase the number of movements detected, providing a better feeling of representation, connection, and control of the game in real-time [10]. The cited studies found statistically significant improvements in strength and dexterity of the upper limb, as well as high user satisfaction. On the other hand, Sattar et al. [20] combined the MYO Armaband^®^ with fNIRS to increase the accuracy to obtain information about muscle activity and motor intention, achieving an accuracy of over 90%. These findings suggest that using the MYO Armband^®^ in combination with other VR devices could improve data collection regarding muscle activity, movement, usability, and interaction, which could translate into better clinical effects. Sattar et al., also examined the use of these two devices combined to collect data about motor intention in order to improve control of upper limb prostheses in transhumeral amputees. While the MYO Armband^®^ predicted flexion, extension, pronation, and supination of the elbow, the fNIRS obtained signals for hand opening and closing. For this reason, this approach can be useful to enhance the control performances of multifunctional prostheses with a high level of accuracy. However, and even though this is an innovative approach since it is the first time that someone combines these devices pursuing this goal, further research is required to learn if this strategy is more efficient, comfortable, and beneficial than the ones currently implemented.

In regard to the data concerning the accuracy of the device, although the articles examined reported high percentages of the accuracy of the sEMG system of the MYO Armband^®^ [4,11,18,19,20,21], these results cannot be generalized due to their small sample size. Only one study analyzed the operating time [4], reporting positive results on the speed of the system to detect gestures in upper limb amputees, and opening the possibility of introducing more complex movements in future research. This study showed that the average time for a gesture to be detected was 2.62 s, which the authors considered suitable for gaming. Additionally, three studies analyzed the accuracy of the device to obtain information about electromyography, orientation, and position in people with stroke, finding heterogeneous results. Lyu et al. [18] found an accuracy below 92.5%, which was the accuracy found in a sample of healthy subjects; although they did not report any problem regarding calibration, donning, or executing tasks with the device. Ryser et al. [11] found an accuracy of 78–99% in a sample of people with mildly to severely impaired hand function, whereas Castiblanco et al. [21] reported an accuracy of 85%. These results should be interpreted with caution, since the sample size was very small (n = 9), and the participants presented different levels of disability and upper limb impairment. We observe the same situation when examining the articles by Gaetani et al. [19] and Sattar et al. [20]: they reported an accuracy over 73% in transhumeral amputees and over 93% in transradial amputees, with a response time below 1 s, but their sample size was very small in both studies (n = 1 and n = 4 respectively). Anyhow, an accuracy over 75% provides an argument in favor of the MYO Armband^®^.

Regarding the studies focused on the clinical effects of interventions with the MYO Armband^®^, those with samples of people with stroke and MS found improvements in upper limb ROM and function [7,9,10]. These results are consistent with the meta-analysis published by Cortés-Pérez et al. [29] who reported that the Leap Motion Capture System (LMCS), another semi-immersive VR commercial device, improved upper limb function and ROM in people with neurological disease. They also observed that these clinical effects were higher when the VR-based therapy was combined with conventional therapy. In addition, Avcil et al. [30] conducted a randomized clinical trial and found that video game-based therapy using Nintendo^®^ Wii and LMCS enhanced significantly dexterity, grip strength, and functional ability in people with CP. In our study, the only article that included a sample with CP [22] also found significant changes in grip strength, bimanual activities, dexterity, function, functional performance, and participation. However, we must note that, in our view, the heterogeneity of the interventions, the small sample size, the short duration of the protocols, and the lack of a control group may have diminished the effect of the interventions.

The dosage of therapy was very heterogeneous between studies. While the study by Esfahlani et al. [10] on people with MS implemented a 10-week protocol with 1-h sessions five days per week, the same research team implemented an 8-week protocol with 1-h sessions in a different study with people with stroke, although in this case, they did not specify the number of sessions per week [7]. Lamers et al. [31] concluded that there is no consensus on the optimum dosage of upper limb rehabilitation in people with MS. However, most of the studies included in their systematic review had an intervention duration of 8 weeks or more, with 30–60-min sessions 2–5 days per week. Also, it should be noted that the systematic review by Laver et al. [32] published by the Cochrane Library suggested that RV protocols that provided more than 15 h of therapy resulted in greater benefits in people with stroke than those providing a smaller dose.

Another advantage of VR observed in the studies included in this review is that it can improve user’s motivation, interest, adherence, and satisfaction towards therapy [7,9,10], and this is common to other studies that examine semi-immersive VR-based therapy in different disorders, such as cardiovascular diseases [33,34], brain damage [35,36], children and adults with CP [37,38,39], neurodegenerative diseases [40,41] or chronic pain [42].

Only one study included in this review analyzed the adverse effects of the interventions with the MYO Armband^®^, reporting no adverse events during or after therapy [22].

Nevertheless, the risk of bias was high for all the articles, so the results should be interpreted with caution. Only 10% of the studies randomized the sample, in most of them the assessors and/or therapists were not blind and in at least 30% the results were incomplete. Also, the level of evidence and grade of recommendation were low since all studies were case series.

### Limitations

There are some limitations to this review that are important to highlight. First, due to the heterogeneity of the interventions, outcome measures, and dosage, it was impossible to conduct a meta-analysis of the results. Also, we only selected articles published in English or Spanish in the last 5 years and the search was limited to a few databases, which may have reduced the number of articles included. In addition, the low methodological quality of the studies, the small and heterogeneous samples, the high risk of bias, and low level of evidence and grade of recommendation are factors that may limit the extrapolation of our results to all patients with motor impairment of the upper limb.

## 5. Conclusions

VR systems appear to be an effective rehabilitation approach when combined with conventional therapy in people with motor impairment of the upper limb. Specifically, therapy with the MYO Armband^®^ has shown clinical changes in ROM, dexterity, performance, functionality and satisfaction. It has also proven to be an accurate system to capture signals from the forearm muscles in people with motor impairment of the upper limb. However, further research should be conducted using bigger samples, well-defined protocols, comparing with control groups or comparing with other assessment or therapeutic tools, since the studies published so far present a high risk of bias and low level of evidence and grade of recommendation.

## Figures and Tables

**Figure 1 jcm-11-03119-f001:**
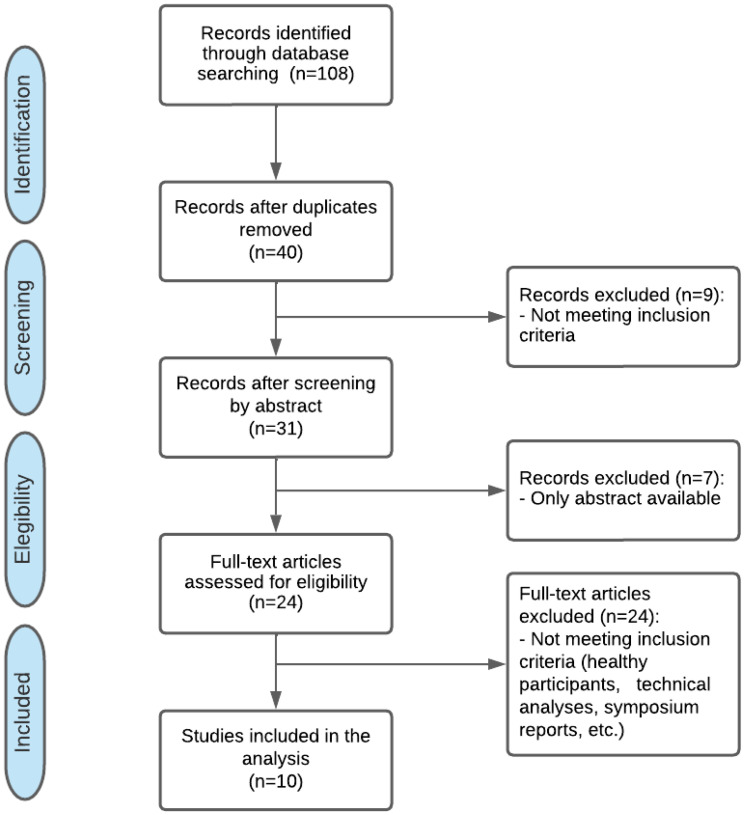
PRISMA Flow chart for identifying studies for systematic review.

**Figure 2 jcm-11-03119-f002:**
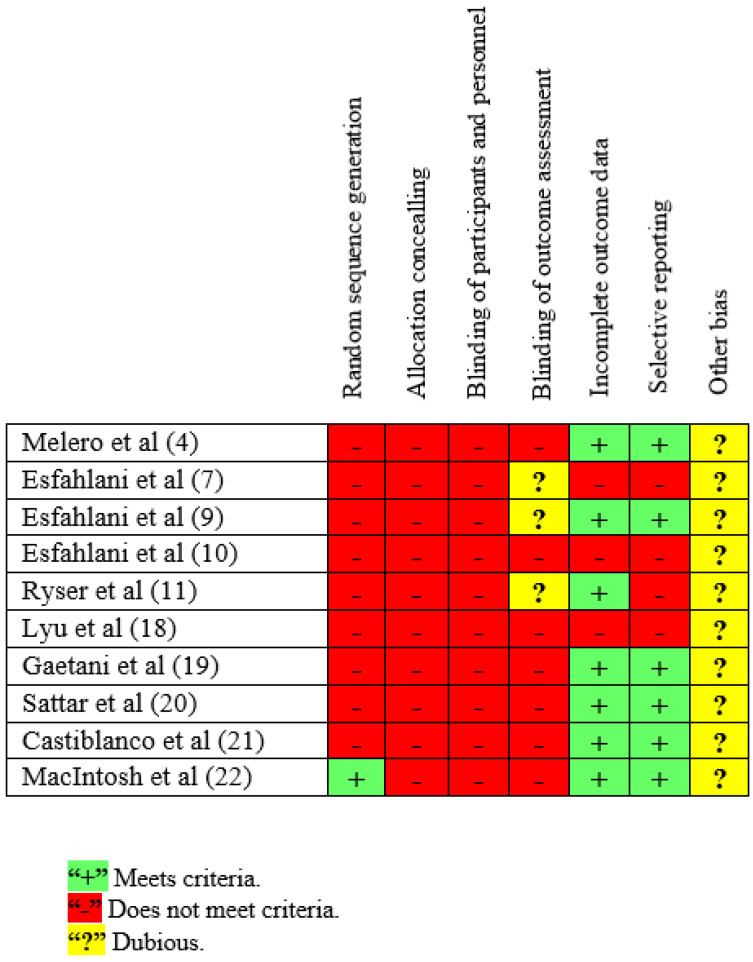
Assessment of risk of bias. Assessments by the reviewers for each risk sorted by article.

**Table 1 jcm-11-03119-t001:** Search filters in databases.

Database	Search Filter
*PubMed*	- Availability: full text - Publication date: last 5 years
*Scopus*	- Year: 2016, 2017, 2018, 2019, 2020, 2021 - Language: English - Document type: any article
*Web of Science*	- Year: 2016, 2017, 2018, 2019, 2020, 2021 - Language: English - Document type: any article
*ScienceDirect*	- Year: 2016, 2017, 2018, 2019, 2020, 2021 - Language: English - Article type: research article - Subject area: engineering, computer science, neuroscience
*PEDro*	No filter
*Cochrane Library*	- Year: from 2016 until 2021

**Table 2 jcm-11-03119-t002:** Characteristics of the studies focused on the accuracy of the device as an assessment tool.

Study	Participants (Disease and Sample Size)	Protocol	Data Collection Process	Outcome Measures	Results
Melero et al. [4]	Disease: amputation n: 3	Dance game with visual feedback using Kinect^®^ + MYO Armband^®^	10 game trials for each patient	Detection time, reaction time and operating time. O = R + D	D = 0.24 s/R = 0.92 s/O = 1.15 s MD = 2.6 s Operating time (R + MD) = 3.56 s Initial expected operating time = 6 s
Ryser et al. [11]	Disease: stroke n: 3	Assessing the accuracy of the MYO^®^ to detect movement intention in order to control a dynamic hand orthosis device	Performing three gestures, each for 60 s	Classification algorithm	Accuracy for five gestures for all samples: 98%, Accuracy for three gestures related to ADL: 94.3% Accuracy in people with stroke: 78–99%. The system is suitable for stroke rehabilitation
Lyu et al. [18]	Disease: stroke n: 6 healthy + 2 stroke	Visuomotor training task using the MYO Armband^®^	Accuracy: four gestures, 25 repetitions per gesture, 4 s contraction, 2 s relaxation, 30 s rest Validation: 36 blocks of exercises, four trials per block	sEMG signals captured via MYO^®^	Accuracy: 99.3% for wrist extension, 82% for radial deviation, 100% for flexion Accuracy in healthy subjects: 92.5% Validation: task performance improves through training Stroke patients: no event was reported regarding calibration, donning, or executing tasks with the device. Lower accuracy than healthy subjects
Gaetani et al. [19]	Disease: transradial amputation n: 9 (8 healthy + 1 congenital amelia)	performing three different gestures with hand fingers to collect sEMG data with the MYO^®^ and analyze accuracy and response time	10 s of flexion, 10 s of extension, and 10 s of rest	Learning algorithm, analysis of sEMG signal	Average accuracy of gesture recognition: 90.4% Accuracy in subject with amputation: 93.3% Response time: <1 s The system works also on subjects with small not-trained muscles
Sattar et al. [20]	Disease: transhumeral amputation n: 18 (15 healthy + 3 amputees)	Creation of BCI to control upper limb prostheses: sEMG (MYO Armband^®^) + fNIRS The armband acquired the sEMG signals for four-arm motions: elbow extension, elbow flexion, wrist pronation, and wrist supination	Training session: resting period of 3 min to establish a data baseline. Data acquisition: initial 5-sec rest followed by a 20-sec task period	Data processing from sEMG and fNIRS using MATLAB^®^	The hybrid sEMG and fNIRS system is a feasible approach to improve the CA for transhumeral amputees, improving the control performances of multifunctional upper-limb prostheses. The average accuracy of 94.6% and 74% was achieved for elbow and wrist motions by sEMG for healthy and amputated subjects, respectively Simultaneously, the fNIRS modality showed an average accuracy of 96.9% and 94.5% for hand motions of healthy and amputated subjects
Castiblanco et al. [21]	Disease: stroke n: 10 (6 healthy + 4 stroke)	Healthy: collection of six sEMG signals (four from right arm and two from left). One trial. Stroke: 12 sEMG signals (eight from impaired side, two from non-impaired). Three trials. All with visual feedback	Maintaining each movement 3–5 s (open-close the hand, flexion-extension of the wrist, spread the fingers, and pinch-grip each finger)	Classification algorithms	Exercises with best performance: opening-closing hand Exercises with worst performance: pinch-grip finger it was possible to identify the hand movements from sEMG signals for subjects who had a motor disability due to stroke with a correct classification rate of 85%

ADL: activities of daily living; BCI: Brain-computer interface; D: detection time; fNIRS: Functional near-infrared spectroscopy; MD: maximum detection time; O: operating time; R: reaction time; sEMG: Surface electromyography.

**Table 3 jcm-11-03119-t003:** Characteristics of the studies focused on the clinical effects of the device as a rehabilitation tool.

Study	Participants (Disease and Sample Size)	Intervention or Protocol	Dosage	Outcome Measures	Results
Esfahlani et al. [7]	Disease: stroke n: 20	3D games controlled with Kinect^®^ and MYO^®^	8 weeks 1 h/day, (days per week not specified)	EQ (Rasch Analysis), MAS, angular velocity, acceleration, ROM	Flow, presence, and absorption EQ: participants enjoyed the sessions the activities covered a good ROM for the upper body Suggest audio feedback
Esfahlani et al. [9]	Disease: stroke, MA and TBI n: 23 (10 healthy CG; 2 stroke, 2 TBI and 9 MA IG)	Serious game controlled by Kinect^®^ + MYO^®^ + pedal	45-minute sessions, no further information	ROM response time, electromyographic data, velocity, orientation, and inertial information	Improvement in performance reflected in response time and ROM High interest and engagement The combination of MYO^®^ and Kinect^®^ increase the accuracy to detect gestures
Esfahlani et al. [10]	Disease: MS n: 52 (40 MS IG; 12 healthy CG)	IG: video games using Kinect + MYO + Pedal GC: not specified	10 weeks 5 days/week 1 h/day	MAS, ROM	Statistically significant differences in performance and ROM. High interest and engagement
MacIntosh et al. [22]	Disease: CP n: 19	Video game controlled by completing therapeutic gestures detected via electromyography and inertial sensors on the forearm via the MYO^®^ and custom software	4 weeks 17 min/day	AHA, BBT, wrist extension, grip strength, COPM, SEAS	Moderate improvements in active writs extension, grip strength, COPM and BBT, small improvement in AHA Positive results in SEAS No adverse effects

AHA: Assisting Hand Assessment; BBT: Box & Blocks Test; COPM: Canadian Occupational Performance Measure; CP: Cerebral Palsy; EQ: Engagement Questionnaire; MS: multiple sclerosis, CG: control group; IG: intervention group; TBI: traumatic brain injury; ROM: range of movement; SEAS: Self-Reported Experiences of Activity Settings.

**Table 4 jcm-11-03119-t004:** Scores for each article after evaluation with the tool for the critical appraisal of epidemiological cross-sectional studies.

	Tool item	Melero et al. [4]	Esfahlani et al. [7]	Esfahlani et al. [9]	Esfahlani et al. [10]	Ryser et al. [11]	Lyu et al. [18]	Gaetani et al. [19]	Sattar et al. [20]	Castiblanco et al. [21]	MacIntosh et al. [22]
Methodological analysis	1	VG	F	F	F	F	VG	F	VG	VG	VG
	2	NS	B	B	NS	B	NS	B	F	B	G
	3	NS	B	B	F	B	NS	B	F	F	G
	4	NS	B	B	G	B	B	B	B	B	F
	5	NS	NS	B	NS	B	NS	B	B	B	G
	6	B	NS	NS	NS	B	G	F	NS	F	VG
	7	NA	NA	B	G	NA	F	NA	B	F	G
	8	NA	NA	F	B	NA	B	NA	G	F	G
	9	NA	NA	G	G	NA	B	NA	G	B	G
	10	NA	NA	NS	NS	NA	F	NA	NS	NS	G
	11	G	G	F	G	F	F	F	F	F	VG
	12	F	G	B	F	F	F	NS	F	F	VG
	13	NS	G	B	F	B	NS	G	F	NS	VG
	14	B	F	F	F	F	F	B	F	F	G
	15	F	B	VG	F	NS	F	B	B	G	F
	16	F	F	G	G	NS	G	NS	B	G	F
	17	G	NS	NS	NS	NS	NS	NS	B	NS	NS
	18	F	NS	NS	NS	NS	F	G	B	NS	NS
	19	VG	B	F	F	F	F	F	G	G	G
	20	F	F	G	G	G	G	F	G	G	VG
	21	F	B	F	G	F	F	B	G	G	VG
	22	NA	B	VG	G	B	F	G	NS	F	G
	23	G	F	G	G	VG	G	G	F	NS	G
	24	G	F	G	VG	VG	G	B	F	NS	G
	25	B	B	B	B	B	B	B	B	B	F
	26	NS	B	F	B	F	F	B	B	NS	B
	27	VG	VG	VG	NS	B	NS	F	NS	F	F
	28	NS	G	NS	G	NS	NS	NS	NS	NS	VG
	29	NS	NS	NS	NS	NS	NS	NS	NS	NS	VG
	30	NS	NS	NS	NS	NS	NS	NS	NS	NS	VG
	31	NS	NS	NS	NS	NS	NS	NS	NS	NS	NS
	Internal validity	LOW	LOW	LOW	MEDIUM	LOW	MEDIUM	LOW	LOW	MEDIUM	HIGH
	External validity	LOW	LOW	LOW	LOW	LOW	LOW	LOW	LOW	LOW	LOW
	Overall quality	LOW	LOW	LOW	MEDIUM	LOW	MEDIUM	LOW	LOW	MEDIUM	HIGH

VG: very good; G: good; F: fair; B: bad; NA: not applicable; NS: not specified.

**Table 5 jcm-11-03119-t005:** Levels of evidence and grades of recommendation established by the Oxford Center for Evidence-based Medicine.

Study	Level of Evidence	Grade of Recommendation
Melero et al. [4]	4	C
Esfahlani et al. [7]	4	C
Esfahlani et al. [9]	4	C
Esfahlani et al. [10]	4	C
Ryser et al. [11]	4	C
Lyu et al. [18]	4	C
Gaetani et al. [19]	4	C
Sattar et al. [20]	4	C
Castiblanco et al. [21]	4	C
MacIntosh et al. [22]	4	C

## Data Availability

Not applicable.

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
