# Peer review of "An sEMG-Controlled Forearm Bracelet for Assessing and Training Manual Dexterity in Rehabilitation: A Systematic Review"

_jcm, 2022, doi:10.3390/jcm11113119_

Round 1
Reviewer 1 Report
In this paper, the authors have systematically reviewed the use of MYO Armband for assessing and training manual dexterity in rehabilitation. The paper is well-organized and well-written and the search strategy and scientific content seem sound. There are few points that should be revised.
- In the Title of the paper, "a sEMG" should be corrected as "an sEMG".
- Since few (just 10) papers have met the criteria to be reviewed, why just the papers after 2016 were considered in this review? Isn't there any paper published before 2016 in this field?
- In Table 1, 20217 should be 2021.
- The fonts in Figure 1 are low-resolution and should be enhanced.
- In Table 2, "all sample" should be "all samples".
- In Table 2, the parenthesis ")" after fNIRS should be removed.
- In Table 2, some spaces seems unnecessary and should be removed, e.g. those within the following parts: "resting period of 3 mins", "5-sec rest followed", etc.
- In Table 2, what do you mean by "y" in "sEMG y fNIRS"?
- In Table 2, please specify the name of software in "sEMG y fNIRS using software".
- In the caption of Table 3, there are two studies as "studies studies".
- Please write "cerebral palsy" with capital first letters as "Cerebral Palsy" where CP is defined below Table 3.
- in Line 288, "an accuracy for of" seems incorrect. I think you mean "an accuracy of".
- In Table 4, I don't understand what the numbers 1 to 31 stand for?
- Figure 2 seems redundant since Figure 3 covers its content completely. If so, please remove Figure 2.
- Please carefully go through the manuscript to correct a few grammatical errors.
Author Response
AN SEMG-CONTROLLED FOREARM BRACELET FOR ASSESSING AND TRAINING MANUAL DEXTERITY IN REHABILITATION: A SYSTEMATIC REVIEW
Reviewer reports:
Thank you for the opportunity to revise our manuscript. We appreciate your comments and constructive suggestions. It is our belief that the resubmitted manuscript has improved after making the suggested edits.
Revisions in the text are shown in a marked-up copy using yellow highlight for additions and edits. The revision, based on the review team’s collective input, includes several positive changes.
Reviewer 1:
Comments and Suggestions for Authors
In this paper, the authors have systematically reviewed the use of MYO Armband for assessing and training manual dexterity in rehabilitation. The paper is well-organized and well-written and the search strategy and scientific content seem sound. There are few points that should be revised.
--> Thank you for the opportunity to revise our manuscript. We appreciate your comments and constructive suggestions. It is our belief that the resubmitted manuscript has improved after making the suggested edits.
- In the Title of the paper, "a sEMG" should be corrected as "an sEMG".
Thank you for this correction. Done (page 1).
- Since few (just 10) papers have met the criteria to be reviewed, why just the papers after 2016 were considered in this review? Isn't there any paper published before 2016 in this field?
Thank you for your comment. We did not include articles prior to 2016 for two reasons: first of all, the device was launched by Thalmic Labs in 2014, so there are not articles before that date. Secondly, there are not articles analyzing the accuracy and/or the clinical effects of using the MYO Armband® in people with motor impairment of the upper extremity before 2016 which met the inclusion criteria.
- In Table 1, 20217 should be 2021.
Thank you for the correction. Done (page 3).
- The fonts in Figure 1 are low-resolution and should be enhanced.
Thank you for the correction. Done (page 5).
- In Table 2, "all sample" should be "all samples".
Thank you for the correction. Done (page 6).
- In Table 2, the parenthesis ")" after fNIRS should be removed.
Thank you for the correction. Done (page 7).
- In Table 2, some spaces seems unnecessary and should be removed, e.g. those within the following parts: "resting period of 3 mins", "5-sec rest followed", etc.
Thank you for the correction. Done (page 7).
- In Table 2, what do you mean by "y" in "sEMG y fNIRS"?
Thank you for the correction. Sorry, it was a translation error. Done (page 7).
- In Table 2, please specify the name of software in "sEMG y fNIRS using software".
Thank you for the correction. Done (page 7).
- In the caption of Table 3, there are two studies as "studies studies".
Thank you for the correction. The same also happened in Table 2. Done (pages 6 and 9).
- Please write "cerebral palsy" with capital first letters as "Cerebral Palsy" where CP is defined below Table 3.
Done (page 10).
- in Line 288, "an accuracy for of" seems incorrect. I think you mean "an accuracy of".
Thank you for the correction.
- In Table 4, I don't understand what the numbers 1 to 31 stand for?
Thank you for the correction. Sorry, we forgot to indicate it. It represents the 31 tool items for the critical appraisal of epidemiological cross-sectional studies. Done (page 13).
- Figure 2 seems redundant since Figure 3 covers its content completely. If so, please remove Figure 2.
Thank you for the suggestion. We have improved the figure resolution on page 14 and modified the text on page 12.
- Please carefully go through the manuscript to correct a few grammatical errors.
Done.
Thank you for your recommendations.
Sincerely yours,
The Authors

Reviewer 2 Report
The authors presented a very well written review. They described in details the methodolgy used for this systematic review.
The topic is very interesting and the results of the study are helpful for future uses of MYO Armband in research.
Author Response
AN SEMG-CONTROLLED FOREARM BRACELET FOR ASSESSING AND TRAINING MANUAL DEXTERITY IN REHABILITATION: A SYSTEMATIC REVIEW
Reviewer reports:
Thank you for the opportunity to revise our manuscript. We appreciate your comments and constructive suggestions. It is our belief that the resubmitted manuscript has improved after making the suggested edits.
Revisions in the text are shown in a marked-up copy using yellow highlight for additions and edits. The revision, based on the review team’s collective input, includes several positive changes.
Reviewer 2:
The authors presented a very well written review. They described in detail the methodology used for this systematic review.
The topic is very interesting, and the results of the study are helpful for future uses of MYO Armband in research.
--> Thank you for the opportunity to revise our manuscript. We appreciate your comments and constructive suggestions. It is our belief that the resubmitted manuscript has improved after making the suggested edits.
Thank you for all your comments.
Sincerely yours,
The Authors
